# Tall Fescue (*Festuca arundinacea* Schreb.) Shows Intraspecific Variability in Response to Temperature during Germination

**Lina Q. Ahmed and Abraham J. Escobar-Gutiérrez ***

INRAE, URP3F, Le Chêne-BP 6, F-86600 Lusignan, France; ahmedlin.inra16@gmail.com
* Correspondence: abraham.escobar-gutierrez@inrae.fr; Tel.:+33-549556184; Fax: +33-549556068

**Abstract:** Tall fescue is a major species growing in temperate grasslands. It is a cool-season perennial native of Western Europe and used worldwide as forage for its quality and adaptability to various soils and climates. By its effects on germination and seedling growth, temperature affects the recruitment of individuals and, consequently, the genetic diversity of plant communities. Under most climate change scenarios, breeding cultivars adapted to new ranges of temperature will be necessary. Knowing the variability of the responses to temperature by different accessions is an essential first step towards such breeding. In this work, we (i) analyze the intraspecific variability of tall fescue in response to a constant temperature during germination and (ii) quantitatively describe the response curves. A sample of nine, from 128, accessions of tall fescue was evaluated. Four replicates of 100 seeds per accessions were tested for germination in the dark at eight constant temperatures ranging from 5 to 40 °C with increases of 5 °C. The germinability, lag to start and maximum germination rates were estimated. It was observed that the responses of tall fescue accessions were statistically different ($p < 0.05$). The optimal temperature for maximum germination ranged from 9 °C to 25 °C. Germination was not observed for any accession at 40 °C. The novelty of this work comes from the duration of the sampling period at low temperatures that was longer than in most published papers. Based on the responses to temperature during germination, our findings suggest that a high intraspecific genetic variability exists in tall fescue that merits further exploration. This variability should be useful to breed new cultivars adapted to the new environmental conditions imposed by the ongoing fast climate change.

**Keywords:** breeding; forage; genetic diversity; germinability; grassland

## 1. Introduction

Nowadays, it is possible to predict the degree of success of a species based on the capacity of their seeds to spread and germinate through time [1,2]. Germination is the first crucial stage for plant establishment. It is a complex adaptive trait of higher plants, and it is influenced by environmental factors and the genetic patrimony of the seed [3–5]. Understanding germination responses to environmental factors and the underpinning genetic determinants is important for physiologists, molecular biologists, seed technologists and ecologists.

By its effects on germination and seedling growth, temperature affects the recruitment of individuals and, consequently, the genetic diversity of plant communities. The rates of chemical and biochemical reactions, as well as the rates of cell growth and plant functioning, all depend on the temperature [6,7]. The relationship between seed germination and temperature has been extensively studied [2,8], and several mathematical models have been proposed to describe such a relationship that can be expressed in terms of the cardinal temperatures [9–12].

Tall fescue (*Festuca arundinacea* Schreb.), is a perennial and caespitose cool-season grass [13]. In the Poaceae family, the *Festuca* genus is one of the largest, with more than 500 species [14]. *F. arundinacea* can be used flexibly in various farming systems; it can

be cultivated in association with legumes (*Trifolium repens* and *T. pratense*) or with other grasses for hay or silage production [15]. Although Mediterranean varieties are sensitive to low temperatures [13], tall fescue from temperate regions is well-adapted to climatic extremes of heat, drought and cold [16,17]. Thus, in temperate regions, *F. arundinacea* is being increasingly cultivated due to the impact of progressive climatic changes [14]. Further, in such areas, it is also increasingly used as a turf species because of its heat and drought resistance compared with other perennial cool-seasons grasses, such as *Lolium perenne* and *Poa pratensis* L. [13,15].

The analysis of intraspecific genetic diversity is crucial for its conservation [5] and utilization [12]. However, as for many other species, published knowledge is scarce concerning the intraspecific variability of tall fescue in response to climate factors, particularly to temperature during germination and early growth [7]. Despite numerous studies having explored the relation between temperature and germination in gramineous grasslands [18–24], there have only been a few reports on within-species genetic diversity [5,6,9,25,26]. For instance, the germination percentage was shown to display very different temperature responses for cocksfoot accessions [9,27,28].

The seed germination response to temperature can be used as an early marker for the selection of genotypes adapted to future climatic conditions. Recently, it has been shown that germination responses to temperature vary depending on the origin of accessions and can be an indicator of the degree of domestication in *Medicago sativa* L. [5,25]. The objective of this work was twofold: (i) analyze the intraspecific variability of tall fescue in response to a constant temperature during germination and (ii) quantitatively describe the response curves

## 2. Materials and Methods

### 2.1. Plant Material

A sample of nine, from 128, accessions of *F. arundinacea* was evaluated. Seven of them were wild populations collected in different places in Morocco (3), France (2), Portugal (1) and Tunisia (1). Two of the nine were commercial cultivars sold in France (Table 1). Seeds were obtained from the Centre de Ressources Génétiques des Espèces Fourragères (INRAE-URP3F, Lusignan, France: 46°24'15" N, 0°04'45" E, http://florilege.arcad-project.org/fr/collections/collection-especes-fourrageres-a-gazon, accessed 22 May 2022). Seed were stored at 5 °C and 30% relative humidity in opaque envelopes in the dark until they were used. The seed dry weight (SDW) was determined before starting experiments for four replicates of 100 seeds for each accession.

**Table 1.** Information of the accessions of *Festuca arundinacea* Schreb. used in this study (http://florilege.arcad-project.org/fr/collections/collection-especes-fourrageres-a-gazon, accessed 22 May 2022).

| Accession | Taxon | Collection Site | Latitude and Longitude | Altitude Above Sea Level (m) | Mean Temperature Warmest Quarter (°C) | Mean Temperature Coldest Quarter (°C) | Precipitation Warmest Quarter (mm) | Precipitation Coldest Quarter (mm) |
|---|---|---|---|---|---|---|---|---|
| FE7045 | subsp. arundinacea var. genuina Schreb. (6×) | Khenifra, Morocco | 32°56′00.89″ N, 5°39′41.85″ W | 864 | 23.5 | 8.8 | 28 | 224 |
| FE7046 | subsp. arundinacea var. atlantigena sensu Markgr.Dann. (8×) | Idem | Idem | Idem | Idem | Idem | Idem | Idem |
| FE7047 | subsp. arundinacea var. letourneuxiana (10×) | Idem | Idem | Idem | Idem | Idem | Idem | Idem |
| FE893 | subsp. arundinacea var. genuina Schreb. (6×) | Saouaf, Tunisia | 36°13′41.75″ N, 10°10′18.22″ E | 151 | 25.8 | 10.5 | 53 | 179 |
| FE572 | subsp. arundinacea Schreb. | Quinger, France | 47°06′11.26″ N, 5°52′59.65″ E | 341 | 18.4 | 2.2 | 260 | 240 |
| FE191 | subsp. arundinacea Schreb. | Lusignan, France | 46°24′04.96″ N, 0°07′20.84″ E | 129 | 18.3 | 4.4 | 246 | 226 |
| FE4392 | subsp. arundinacea Schreb. | Alcobaça, Portugal | 39°32′55.20″ N, 8°58′46.74″ W | 53 | 20 | 11.2 | 42 | 304 |
| Centurion | subsp. arundinacea Schreb. | NA | 43°39′00.45″ N, 3°51′58.30″ E | 47 | NA | NA | NA | NA |
| Soni | subsp. arundinacea Schreb. | NA | 48°38′42.21″ N, 2°50′08.93″ E | 101 | NA | NA | NA | NA |

## 2.2. Germination Experiments

The protocol was previously described [6]. Briefly, two experiments were conducted as follows. In the main experiment, after cold stratification, four repetitions of one hundred seeds per accession were tested for germination in the dark at eight treatments of constant temperature ranging from 5 and 40 °C, with 5 °C increments (Table 2). In each temperature, the design was a randomized complete block, each block a vented plastic box containing the Petri dishes with 100 seeds. Vented boxes were placed in growth cambers, where the temperature and relative humidity of the useful volume (1.5 m$^3$) were continuously monitored by six thermocouples placed within the vented plastic boxes at different positions around the Petri dishes and logged every 20 s.

**Table 2.** Temperatures, relative humidity and vapor pressure deficit (VPD) measured for germination.

| Temperature Treatment (°C) | Temperature Actual (°C) | Relative Humidity (%) | VPD (kPa) | Sampling Frequency (h) | Duration of the Observe (h) |
|---|---|---|---|---|---|
| 5 | 5.0 | 74 | 0.23 | 48 | 3024 |
| 10 | 9.6 | 84 | 0.20 | 16 | 1584 |
| 15 | 14.3 | 76 | 0.41 | 12 | 900 |
| 20 | 19.2 | 74 | 0.61 | 8 | 1072 |
| 25 | 25.0 | 80 | 0.63 | 8 | 1072 |
| 30 | 30.0 | 50 | 2.12 | 8 | 1008 |
| 35 | 34.2 | 65 | 1.97 | 12 | 756 |
| 40 | 40.0 | 55–100 | 3.32–0.00 | 168 | 1680 |

A seed was counted as germinated when the radicle or the coleoptile protruded out of the seed and was at least 2 mm long [8]. Germinated seeds were removed from the Petri dishes, and water was added at each count to ensure nonlimiting moisture. Petri dishes were randomized on their vented plastic box (block) after each count. Since the germination speed varies with the temperature, germination counts were carried out at variable time intervals (8–168 h) and durations (756–3024 h), which depended on the treatment (Table 2).

In the main experiment, even after several weeks, any seed germinated at 40 °C for any accession. On top, under high temperature and humidity, saprophytic fungi developed that made it impossible to follow the seeds beyond twelve weeks. Consequently, a subsidiary experiment was conducted at 40 °C under aseptic conditions, as described previously [6].

## 2.3. Calculations and Germination Model

As in Ghaleb et al. [5], for each set of 100 seeds in a Petri dish, the cumulated number of germinated seed over time was fitted a nonrectangular hyperbola. From there, the parameters $t_{50\%}$, $\tau$, and $\alpha_{50\%}$ were estimated as described in Ahmed and Escobar-Gutiérrez [6]. Further, for each accession, the data were normalized for its maximum germinability observed in any Petri dish, regardless of the temperature treatment [6]. The four replicates per accession were pooled together and plotted against the temperature [6]. Third-degree polynomials were fitted to each data set. It was the same for variables $tc$ and $\tau$.

The $\alpha_{50\%}$ estimates for each accession were fitted a five-parameters model, drawn from the Beta distribution [29], either Equation (1) or Equation (2):

$$\alpha_{50\%} = \alpha_{50\%max} \cdot \left[\left[\frac{(T_{max} - T)}{(T_{max} - T_{opt})} \cdot \frac{(T - T_{min})}{(T_{opt} - T_{min})}\right]^{\frac{T_{max} - T_{opt}}{T_{opt} - T_{min}}}\right]^{\beta} \tag{1}$$

$$\alpha_{50\%} = \alpha_{50\%max} \cdot \left[\left[\frac{(T_{max} - T)}{(T_{max} - T_{opt})} \cdot \frac{(T - T_{min})}{(T_{opt} - T_{min})}\right]^{\frac{T_{opt} - T_{min}}{T_{max} - T_{opt}}}\right]^{\delta} \tag{2}$$

The parameters are: the three cardinal temperatures, $T_{min}$, $T_{opt}$, and $T_{max}$; the germination rate at $T_{opt}$, $\alpha_{50\%max}$; and the shape parameters $\beta$ and $\delta$. The low boundary, $T_{min}$, was fixed to 0 °C.

### 2.4. Statistical Analyses

Statistical analyses were performed as in Ahmed and Escobar-Gutiérrez [6]. Briefly, sequential pairwise comparisons were performed between the best fit of accession *i* and the raw data of accessions one to nine. These tests were performed for maximum germinability, as well as for the germination rate when half of the seeds have germinated, $\alpha_{50\%}$ [6]. All statistical tests were carried out using R language [30].

## 3. Results

### 3.1. Maximum Germinability

In order to compare the responses of the different accessions, the maximum germinability was normalized for the data observed in any Petri dish for each accession, regardless of the temperature treatment. These normalized data were then extensively analyzed. Germination was not observed for any accessions at 40 °C, even when two different essays were conducted. The temperature and time course response–surface plots, presented in Figure 1, were constructed with the average of four replicates of germination at each temperature (5–35 °C) for each accession. Although germination counts were performed for as long as 3024 h (126 days) at 5 °C, the time axis is limited to 2280 h. The response–surface plots (Figure 1) show clearly that the time course responses to temperature of the nine accessions were contrasted, not only in the normalized cumulative germination percentage but also in the shape of the germinability curves (Figure 2).

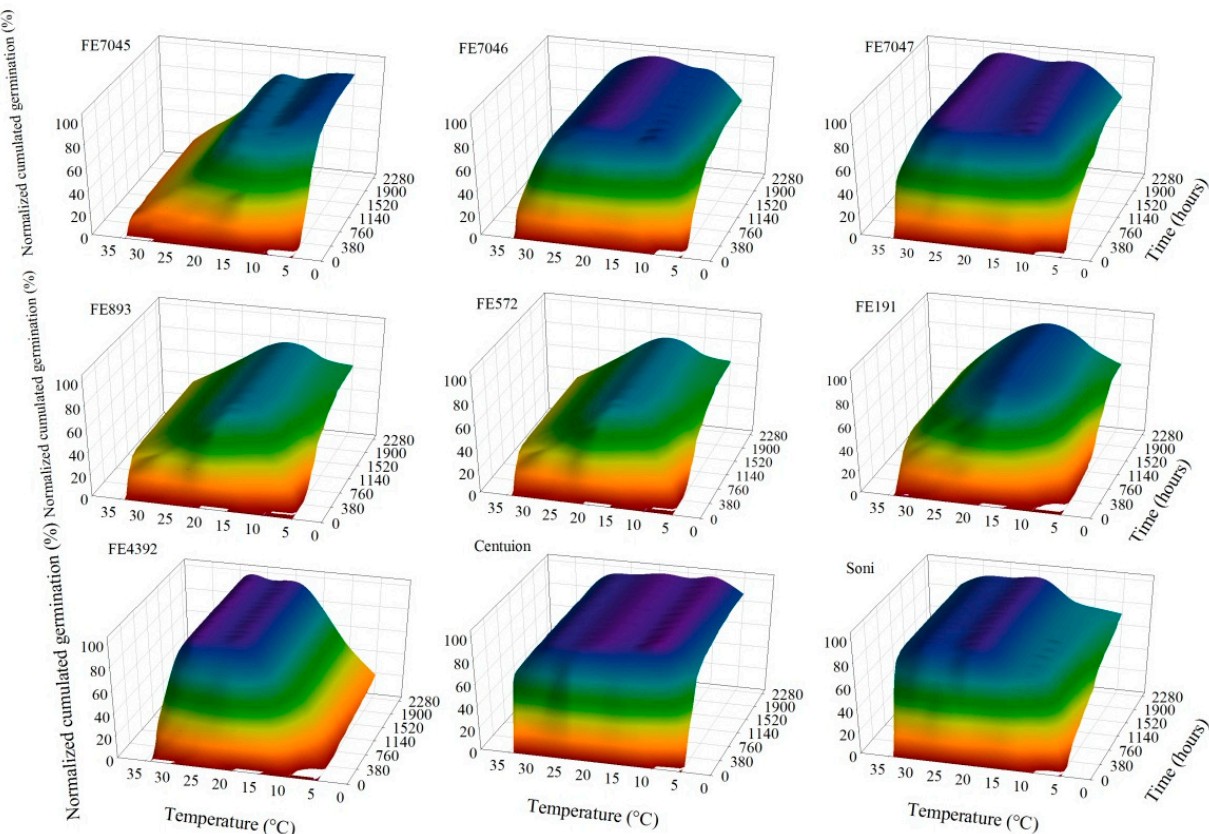

**Figure 1.** Temperature and time course response–surfaces of the normalized cumulative germination percentage of nine accessions of *F. arundinacea*.

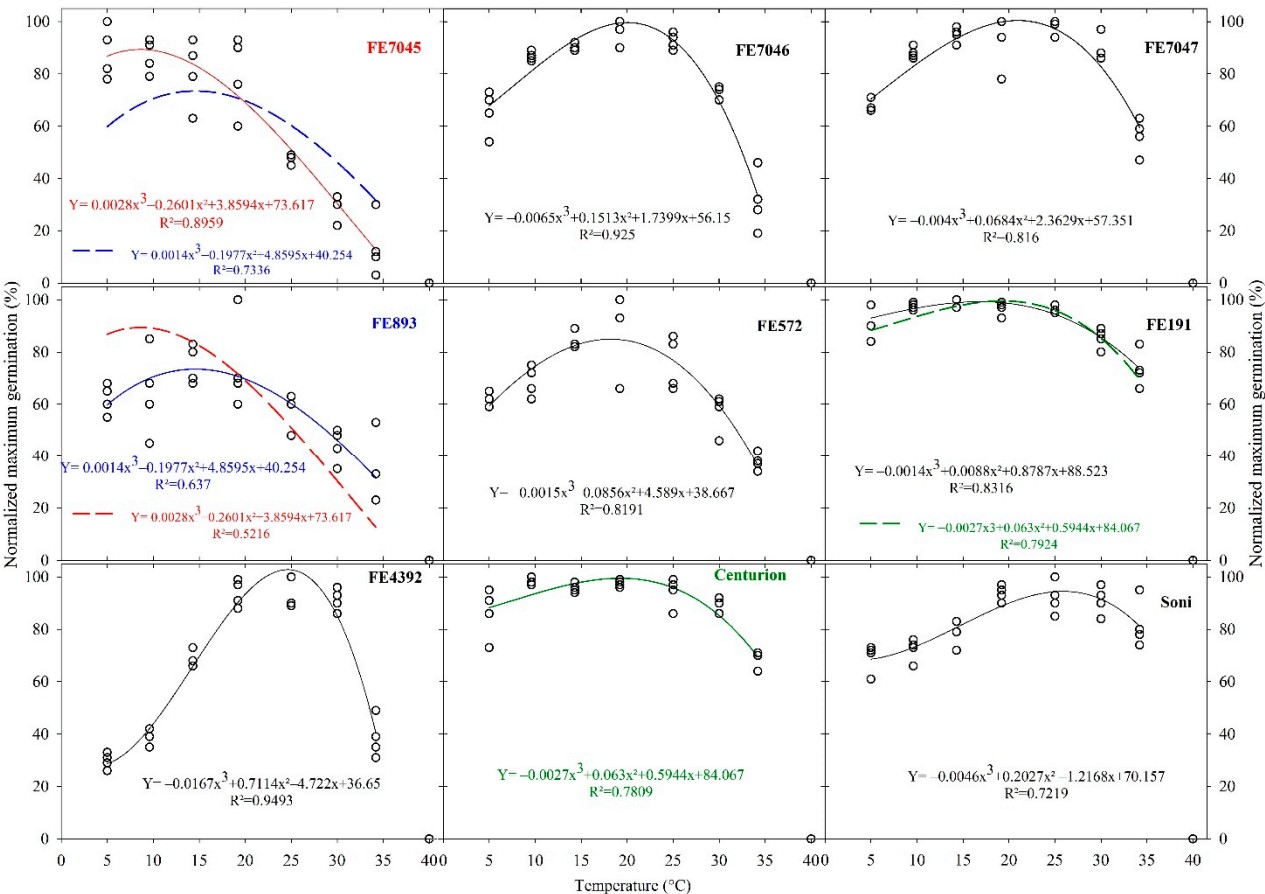

**Figure 2.** Normalized maximum germinability of nine accessions of *F. arundinacea* in response to the constant temperature.

For each accession, the best fitting with a third-degree polynomial is presented in Figure 2. The zero data obtained at 40 °C were excluded from the curve-fitting process.

The pairwise comparisons of normalized maximum germinability curves confirmed the diversity of the genetic material in response to temperature. Indeed, there were only three interchangeable fittings over the forty possibilities (Figure 2).

Only two wild accessions, FE7045 and FE893, have reciprocally exchangeable, statistically valid ($p < 0.05$), adjusting curves. From this, we conclude that they are not different in their response to temperature for seed germination. Further, the second observed similarity concerns wild population FE191 and cultivar Centurion. The caveat here is that the polynomial that best fits Centurion also fits ($p < 0.05$) the data of FE191, but the reciprocal was not true.

Despite the overall diversity in the wide range of temperatures tested, there were some similarities in the responses to particular ranges of temperature. For example, accessions FE7045 and FE893 performed better at temperatures between 5 and 20 °C, while others, such as FE7046, FE7047 and FE572, performed poorly at 5 °C. The behavior of the FE4392 population needs to be highlighted, because of the unique form of the response curve, in which germination increased as the temperature increased from 5 to 25 °C and decreased dramatically after 30 °C. Further, at 5 °C, ca. 3024 h were needed for FE4392 to reach a plateau of cumulated germination that was low (ca. 30%).

For this sample of nine accessions, the optimum temperature for a maximum germinability, estimated from the polynomial fitting, ranged from 8.6 to 25.9 °C.

### 3.2. Time Related Parameters

$tc$ and $\tau$ are parameters of the nonrectangular hyperbola that inform on the time of germination. The responses to constant temperature were significantly different within and between accessions for both parameters (Figure 3).

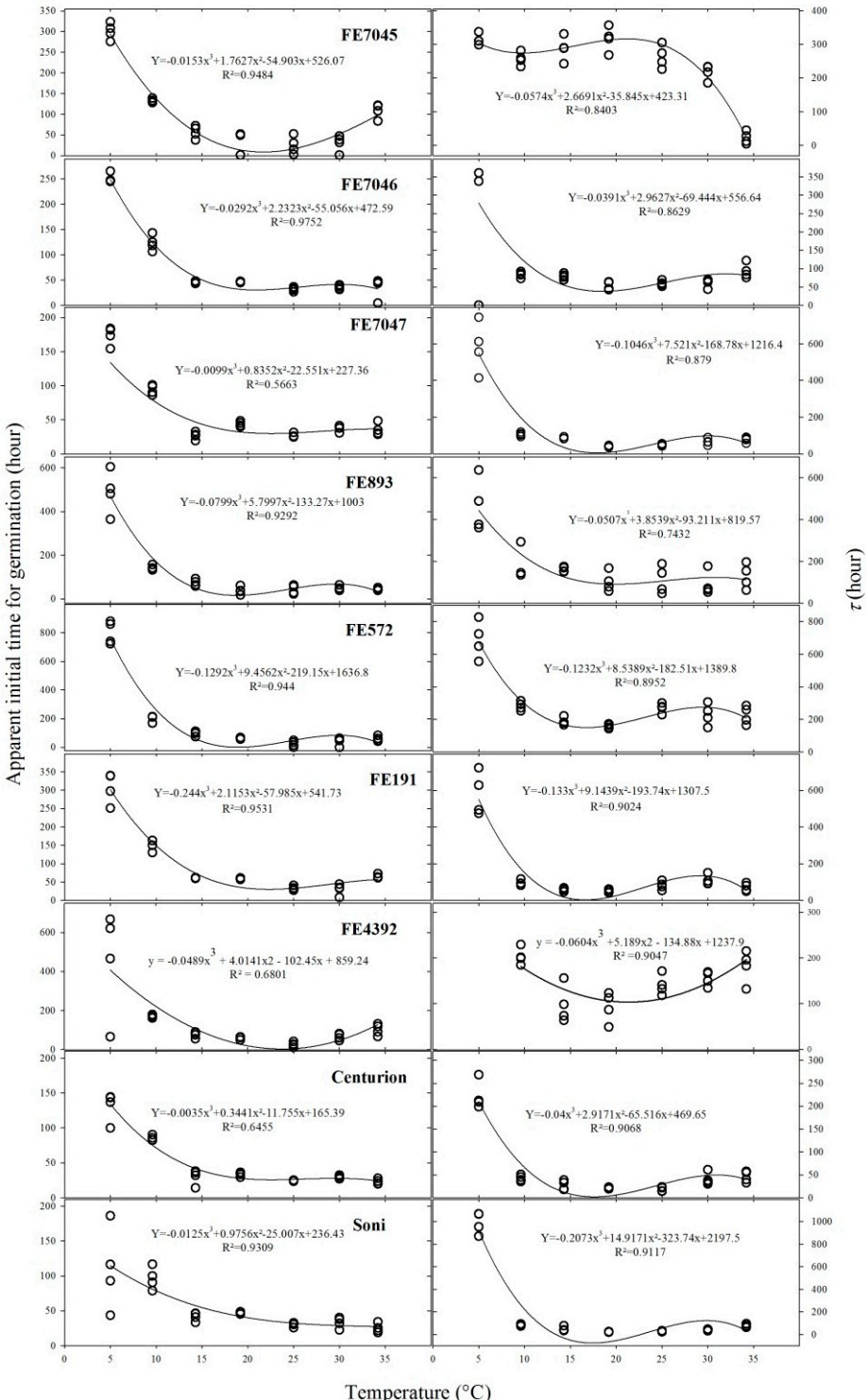

**Figure 3.** Apparent initial time (*tc*) (**Left**) and time when 50% of the seeds germinate ($\tau$) estimated for nine accessions of *F. arundinacea* in response to a constant temperature (**Right**).

Both parameters *tc* and *τ* were greatest at a low temperature (5 °C). Further, parameter *tc* showed a high variability between replicates of cultivar Soni and accession FE4392 germinating at 5 °C (Figure 3, Left).

*τ* showed a similar shape to *tc*, except for population FE7045, which had similar *τ* values between 5 and 25 °C (Figure 3, Right). Further, *τ* values of wild population FE4392 at 5 °C were excluded from the curve fitting, because they were extremely long (ca. 3024 h). Summarizing, accessions FE7047, FE4392, and Soni were more variables at a low temperature (5 °C).

*3.3. Germination Rates*

The germination rates when half of the seeds had germinated (% of seeds per hour), $\alpha_{50\%}$, were fitted to a Beta model (Figure 4).

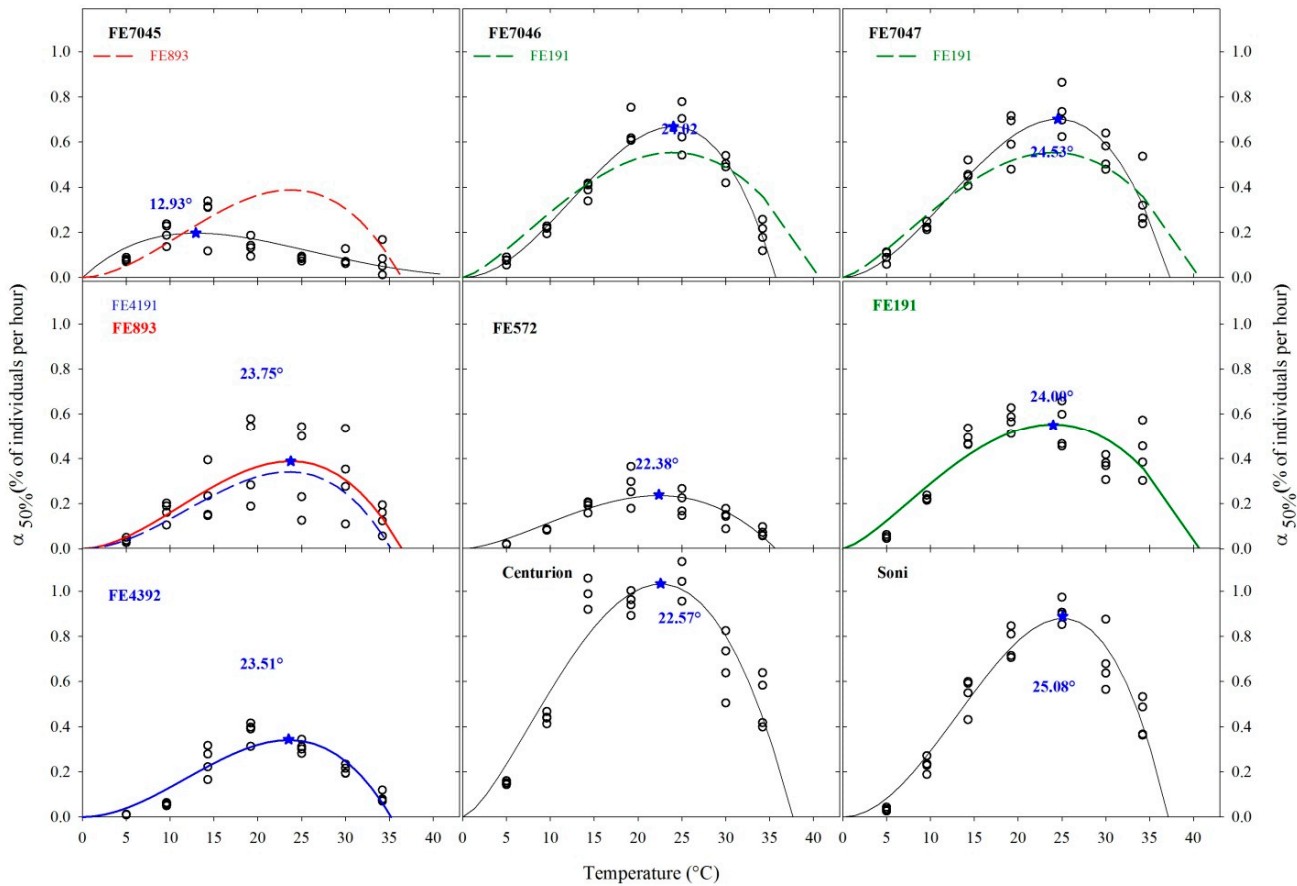

**Figure 4.** Estimated germination rates when 50% of the seeds have germinated ($\alpha_{50\%}$) fitted with the Beta model for nine accessions of *F. arundinacea* in response to a constant temperature. Estimated optimal temperature is indicated by ★.

For curve-fitting purposes, the low boundary was fixed to 0 °C. The upper limit appeared between 35 and 40 °C. The goodness of fit was different between populations. Population FE7045 was fitted the Beta model in the form of Equation (1), because its optimum was found in the first half (12.9 °C) of the experimental temperature range. The other eight accessions, that peaked between 22 and 25.1 °C, were fitted the Beta model in the form of Equation (2). The sequential analysis of pairwise comparisons did not detect reciprocally exchangeable, statistically valid ($p < 0.05$) adjusting curves.

Nevertheless, the curve of wild population FE191 adjusted well the data from accessions FE7046 and FE7047. Likewise, the model of wild population FE7045 was adjusted on the data of wild population FE572, while the model of FE4392 fit the FE893 data. The

responses of the commercial cultivars Centurion and Soni were different ($p < 0.05$) from all the other accessions.

Estimated optimum temperatures were in the range of 22.6–25.1 °C, except for wild population FE7045.

### 3.4. Seed Dry Weight and Size

The nine accessions of *F. arundinacea* had significant differences ($p < 0.05$) in their seed dry weight (SDW) (Figure 5 Left). The SDW of wild population FE572 had the highest value (2.9 mg), whereas cultivar Centurion had the lowest one (1.5 mg). A significant relationship was found ($p < 0.05$) between the average seed dry weight and the non-normalized maximum germination at the optimum temperature (Figure 5 Right). Moreover, linear regression analyses between the seed length and width of the nine accessions (data not shown) revealed a significant ($p < 0.05$) relationship between these two variables only for accessions FE7046, FE893 and FE191.

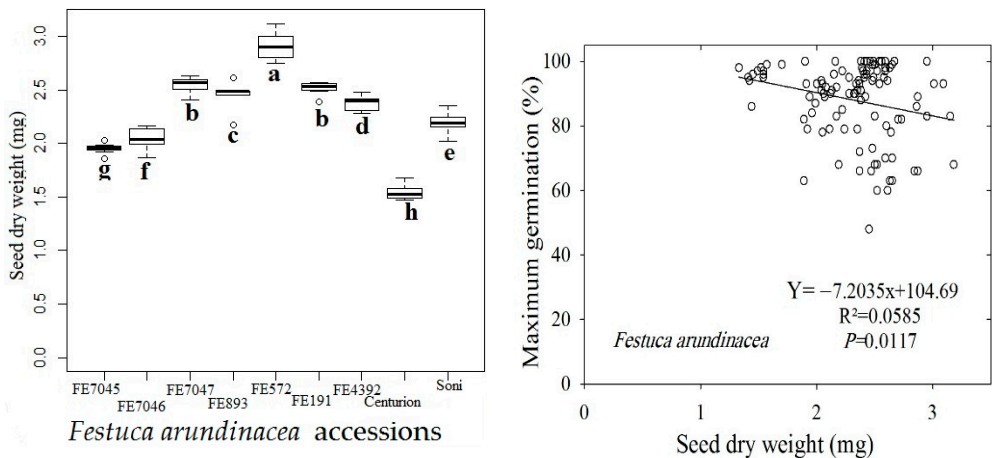

**Figure 5.** Box plots of seed dry weight (SDW) of the nine studied accessions of *F. arundinacea* (**Left**) and regression plot of the maximum germination percentage against SDW (**Right**). Same letter under the boxes indicates that means are not different after a Tukey's HSD test ($p < 0.05$).

## 4. Discussion

Seed germination plays a major role in maintaining the genetic diversity of permanent and cultivated grasslands. In the present research, we evaluated the intraspecific diversity of tall fescue in response to a constant temperature during germination. Although being a topical issue for gene bank curators, breeders and farmers, the subject has received little attention from academics and practitioners. For example, Palazzo and Brar [18] analyzed the interspecific diversity in the germination response to a constant temperature of some species of the genus *Festuca* by using commercial cultivars. Their work included two cultivars of *F. arundinacea*, similarly to Zhang et al. [24] and Monks et al. [23], who also studied germination under constant temperature. Lu et al. [21] evaluated three cultivars under both constant and alternating temperatures. From these four works, it can be concluded that differences exist in the behaviors of commercial cultivars, particularly at the lowest (5 °C) and highest (30–35 °C) temperatures tested. Nevertheless, the length of the observations at these temperatures was, in our opinion, not adapted to the effects of temperature on the rates of biological processes. In our work, the time required to express the full germinability potential of seeds at extremes temperatures was beyond expectations. For example, at 5 °C, while commercial cultivar Centurion reached the plateaus of cumulated germination in ca. 30 days (720 h), Soni needed ca. 95 days (2300 h), and the fastest ecotype, FE7045, needed 40 days (960 h).

Differences also exist in the behavior of accessions at higher temperatures. For example, in our work, some accessions (FE7045 and FE893) germinated less at temperatures over

20 °C. One of the cultivars (Fleche) evaluated by Monks et al. [23] showed similar trends. Further, our accessions from North Africa (FE7045, FE7046 and FE893) appeared sensitive and had low germinability at 30 and 35 °C. These accessions will be better adapted to early spring sowing. Dramatic decreases in germination of tall fescue cultivars were reported at 35 °C [21,23,24] under constant temperature. Indeed, negligible (1%) germination [24] or zero [23] were observed at 35 °C. Negligible (1%) germination was also observed at 37 °C for one of the two cultivars tested by Zhang et al. [24]. As in our work, zero germination at 40 °C was reported by Lu et al. [21] and Zhang et al. [24], which could represent the celling temperature not only for germination but also for many other physiological processes in *F. arundinacea* [7] and other species [5,6,9].

As for other species (e.g., [5,6,9,12,25]), the differences between cultivars discussed above suggested the existence of genetic diversity for the responses to temperature during germination, within the pool of ecotypes of *F. arundinacea* available in gene banks or in nature. In this work, consistent with our hypothesis, intraspecific variability was observed in two cultivars and in a small sample of ecotypes belonging to three subspecies of *F. arundinacea*. This original finding justifies the study of a wider sample of ecotypes that could be a source of genes for breeding. Indeed, the behavior of our wild populations and the recent findings in *Lolium perenne* L. [31] suggest that the current commercial cultivars of tall fescue represent only a small and biased part of the natural diversity of the species *F. arundinacea*.

In an ongoing work, we are analyzing the behavior of over 150 seed lots of *F. arundinacea* from various origins. The future results should confirm the existence of a high intraspecific variability that could be associated with the provenance, as has been shown for *Medicago sativa* L. [5,25]. Potential links between the variability and the ploidy level of accessions also merits being studied.

Temperature had a marked effect on the timing of germination (e.g., $tc$ and $\tau$ in Equation (1)) and the germination rates (% of seeds per hour) such as $\alpha_{50\%}$. From an agronomical perspective, accessions or cultivars presenting low timing lags between imbibition of the dry seed and emergence, as well as a high germination rate, could be interesting for early sowing in order to favor seedling establishment [18,23,24,32].

In addition to the different responses on germinability, our nine accessions showed high diversity for the values of parameters $tc$, $\tau$ and $\alpha_{50\%}$. This is a major finding, because these parameters could be used as early markers for the selection of genotypes adapted to particular environments. For example, a high $tc$ could be interesting for autumn sowings, while small values of $\tau$ could give a competitive advantage in spring sowings. Germination rates reflect the homogeneity of seeds for their timing to germinate. These parameters are also used to determine the cardinal temperatures [23,24]. When considering the rate or speed of germination, $\alpha_{50\%}$, using a nonlinear approach, a base temperature ($T_{min}$) of 0 °C was found for all the accessions of our study, similar to that reported by Monks et al. [23] but in contradiction with the report of Zhang et al. [24]. The range of optimal temperatures ($T_{opt}$) was wide (12.8–25.1 °C) for our nine accessions and rather different from those calculated for the two cultivars by Monks et al. [23]. In coherence with the results of Monks et al. [23], our estimations of $T_{max}$ pointed out that this should be, as discussed above, certainly under 40 °C, as for the other physiological processes in *F. arundinacea* [24].

Concerning the eventual link between SDW and the parameter of germination, we observed differences in the SDW of *F. arundinacea* accessions that could not be related to the ploidy level of accessions. Under alternating temperatures of 5/15 and 15/25 °C, Larsen and Andreasen [33] found a linear relationship between SDW and the mean germination thermal time for *F. rubra*, *Poa pratensis* and *L. perenne*. In our study, such a relationship was not found.

## 5. Conclusions

The literature reporting on the natural diversity of *F. arundinacea* is relatively scarce, despite the importance of this species as a forage resource adapted to a wide variety of environments. Knowledge on the genetic diversity of a species is crucial for any breeding program. In order to go beyond comparisons at particular temperatures, and with the scope of the withdraw general conclusions, we compared the whole response to the range of treatments. Thus, based on the probability of one curve fitting properly the data from another accession, different forms of response were observed, and strong significant differences between accessions were revealed. Indeed, the ecotypes in our sample showed significant difference in their responses. These findings suggest that a high intraspecific genetic variability exists in tall fescue for the response to temperature during germination.

Another novelty of this work comes from the duration of the sampling period at low temperatures, which was longer that in most published papers. On the other hand, no germination at all was observed at 40 °C for any of the nine accessions under study. Thus, the upper temperature limits for germination need to be found. Such high-temperature conditions should be free from saprophytic fungus while maintaining a low vapor pressure deficit.

Individuals or ecotypes that germinate over a relatively wide range of temperatures may be established better in the field than those with a highly specific temperature requirement. The variability we came across, which still needs further exploration on a wider genetic basis, should be useful to breed new cultivars adapted to the new environmental conditions imposed by the ongoing fast climate change.

**Author Contributions:** A.J.E.-G. and L.Q.A. designed the research. L.Q.A. conducted the experiments. A.J.E.-G. and L.Q.A. analyzed the data and wrote the manuscript. All authors have read and agreed to the published version of the manuscript.

**Funding:** This research received no external funding.

**Acknowledgments:** We thank the staff of "Unité de Recherche Pluridisciplinaire Prairies et Plantes Fourragères" (URP3F) for the technical support.

**Conflicts of Interest:** The authors declare no conflict of interest.

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
