# Peer review of "Tall Fescue (Festuca arundinacea Schreb.) Shows Intraspecific Variability in Response to Temperature during Germination"

_agronomy, doi:10.3390/agronomy12051245_

Round 1

Reviewer 1 Report

This study investiaged the effects of temperaturer on seed germination of Tall fescue from 9 accessions and found the suitable tempearature ranges fo 9-25  'C  and the limit temperature of 40  'C for seed germination. The authors condidered that a high intraspecific genetic variability exists in tall fescue. Germinerally speaking, this paper is just a regular manuscript and do not have much novelty. However, it still has some merits. I suggested the publication as a short communication after some revisions.  

  1. In the Abstract section and the Conclusion section, the author consider the one of the novelty of this work comes from the wide range of temperatures and the duration of the sampling period at low temperatures, that was longer that in any published paper. I do not think this is a novelty, in some papers, not only the constant temperatures but also the fluctuating temperatures were used. Authors should make sure the relationship between seed germination and temperatures and points out the ecological meaning of the temperatures.
  2. More and specific discussions should be added about the intraspecific variations in Tall fescue or other species in the Discussion section.
  3. Please do not use so many colours in one Figure i.g. Figure 2. Four colours in one Figure but do not have any use.
  4. Figure 1 is not so much clear for readers and I think it would be better use 2-d Figure to make the point more clear.

Reviewer 2 Report

The authors propose a manuscript titled “Tall fescue (Festuca arundinacea Schreb.) shows intraspecific variability in response to temperature during germination”.

I suggest the following changes:

The aim of the study it is written in the abstract but it is not clear written in the introduction. It is better to be written as the last paragraph in the introduction.

In figure 5 for the SDW the R2 of the function among SDW and maximum germination is very low. It is better to express it with another way may be using box plot or something else.

Author Response

Point 1: The aim of the study it is written in the abstract but it is not clear written in the introduction. It is better to be written as the last paragraph in the introduction.

Response 1: It is done in lines 66-69

Point 2: In figure 5 for the SDW the R2 of the function among SDW and maximum germination is very low. It is better to express it with another way may be using box plot or something else.

Response 2: Indeed, R² is very low.  However, changing the presentation will not change it.  Further, both variables are continuous and not grouped.